**www.cambridge.org/ext**

# Infectious disease as a driver of declines and extinctions

Hamish McCallum[1] (iD), Johannes Foufopoulos[2] and Laura F. Grogan[1]

[1]Centre for Planetary Health and Food Security, Griffith University, Southport, QLD, Australia and [2]School for Environment and Sustainability, University of Michigan, Ann Arbor, MI, USA

**Review**

anthropogenic impacts; wildlife disease; pathogen transmission; infection; declines; extinction

**Corresponding author:**
Hamish McCallum;
Email: H.McCallum@griffith.edu.au

## Abstract

Infectious disease is an important driver of extinctions and population declines. With a few exceptions, such as the fungal disease chytridiomycosis in frogs, disease is probably underestimated as a cause of both local and global extinction because it often co-occurs with other more overt drivers of extinction, and its signs can be easily overlooked. Here, we discuss issues around attributing extinction to infectious disease and overview key underlying factors. We then examine the extent to which anthropogenic influences, such as climate change, habitat destruction and exotic species introductions, are likely to lead to increased extinction risk in association with infectious disease. Finally, we discuss strategies to mitigate the threat of extinction due to infectious disease.

## Impact statement

Infectious disease is increasingly recognised as a major driver of species declines and extinctions. Anthropogenic impacts are likely to lead to an increase in such threats in the future. However, the impact of infectious disease is often cryptic and difficult to recognise, particularly when it acts together with other stressors or when a novel agent is involved. This review is intended to help ecologists recognise disease threats. It also outlines a range of strategies to manage disease threats in declining species.

## Introduction

Extinction is a natural and inevitable process (Lawton and May, 1995). However, the current rate of extinction is several orders of magnitude greater than what has occurred through evolutionary time, and there is little doubt that extinction rates will increase still further in the near future (De Vos et al., 2015). Diamond (1984) listed the "evil quartet" of threatening processes as habitat destruction, overkill, introduction of exotic species and secondary extinctions. Most modern conservation biologists would add climate change to this list. Infectious disease is often not explicitly listed, although it may be associated with each of these, as we will discuss below, and is frequently subsumed as a special case of introduction of exotic species.

Disease can be defined as "an impairment that interferes with… the performance of normal function" (Wobeser, 2007) and is not necessarily caused by an infectious agent. However, here we restrict our attention to infectious diseases – those caused by a biological agent transmitted from one individual to another, either directly, or possibly via other species or via infective stages in the abiotic environment. In addition to viruses and bacteria, infectious biological agents may include fungi, protozoa, metazoan parasites, transmissible cell lines and prions (see Box 1). Due to space constraints, we also focus primarily on diseases that can cause direct additive mortality of vertebrate hosts.

Infectious disease has historically been considered a less important driver of extinction than habitat loss or overexploitation. For example, Smith et al. (2006) found that only 31 of 833 extinctions of animals and plants could be attributed to infectious disease, and Pedersen et al. (2007) found that parasites and pathogens were listed as threatening processes for only 54 mammal species. Figure 1 shows the results of several reviews or meta-analyses that examined infectious disease as a threatening process using the IUCN Red List. However, it is likely that infectious disease has been underrepresented as a threatening process to date due to technological limitations (such as difficulty in detecting pathogens or diagnosing the causal agent for disease), and lack of surveillance and baseline ecological knowledge (Grogan et al., 2014). Also, impacts of disease may be missed because parasites and pathogens can be cryptic, because evidence for underlying mechanism is difficult to find once a species has become extinct, and because many extinctions and endangerments are the result of multiple stressors (Brook et al., 2008). As we will discuss later in this article, there are reasons to expect that extinction risks from parasites and pathogens are increasing and will further increase in the future.

**Box 1.** Glossary. These definitions are taken largely from Foufopoulos et al. (2022).

| Term | Definition |
|---|---|
| Amplification host | A host in which infectious agents multiply to high levels, providing an important source of infection |
| Bridge host | A host (usually other than a vector), which transmits infection from a reservoir host to a focal host, usually via spatial or behavioural overlap. May or may not be an amplification host |
| Focal host | A host species of particular interest, frequently an endangered species. |
| Pathogen | A transmissible biological agent capable of causing disease. Includes viruses, bacteria, fungi, protozoa, metazoan parasites, transmissible cell lines and prions |
| Reservoir host | A host population, species or community in which a parasite or pathogen can be maintained and transmitted to a focal host |
| Resistance (to infection) | The ability to limit parasite burden when exposed to infection |
| Tolerance (of infection) | The ability of a host to limit the harm caused by a given parasite burden |
| Vector | An organism that carries and transmits a pathogen between hosts. Typically small bodied and mobile, often an arthropod |

Unequivocally attributing extinction or decline to infectious disease (or indeed any cause) is not straightforward. It requires close cooperation between a variety of disciplines, including ecologists, pathologists and veterinarians. Given the lack of space, we will not reiterate the approaches that can be taken here, but for further details, see McCallum (2012), Preece et al. (2017), Grogan et al. (2014) and Foufopoulos et al. (2022). One of the very few cases in which the proximate cause of extinction can be attributed unequivocally to infectious disease is that of the Polynesian land snail *Partula turgida.* The death of the last known individuals in a captive colony at the London zoo was caused by a microsporidian infection (Cunningham and Daszak, 1998). However, other factors caused the disappearance of the species in its natural habitat.

The extinction of two species of endemic rodent, *Rattus macleari* and *R. nativitatus*, on Christmas Island in the Indian Ocean (Wyatt et al., 2008) exemplifies some of the difficulties involved in attributing historic extinctions to infectious disease. Black rats, *Rattus rattus*, were introduced to the island in 1899, and within a decade, both endemic species had apparently become extinct. Fortunately, museum specimens of both endemic species are available from before and after the introduction of black rats. The trypanosome *Trypanosoma lewisi* is common in black rats, and PCR techniques enabled detection of trypanosome genetic sequences in the endemic rats after the introduction of black rats, but not before. Coupled with contemporaneous reports of widespread morbidity in endemic rats in the early 1900s, there is a strong indication that those endemic rats may have been driven to extinction by trypanosome infection. Of course, as the endemic rats are extinct, it is not possible to determine experimentally whether trypanosome infection produced high levels of mortality. The ability to even infer a role for infectious disease arises from the unusual circumstances of samples from hosts being available before and after the introduction of the parasite. In almost all other cases of historic extinctions, such samples would not be available, and therefore, any role of infectious disease in the extinction process remains speculative.

However, in many cases, infectious disease can be inferred as the most plausible dominant driver of extinction, or at least an important component cause, without unequivocal "proof". For example, at least 12 species of Hawaiian land birds disappeared around the turn of the 20th century (van Riper et al., 1986), coinciding with the first detection of avian malaria *Plasmodium relictum* on the islands (the main vector of which, *Culex quinquefasciatus,* arrived as early as 1826). Although testing the susceptibility of now extinct birds to malaria is impossible, several extant species persist only at high altitudes, which are too cold for malaria transmission (van Riper et al., 1986). In addition, avian pox (a virus) spread by the same species of mosquito vector has had additional major effects on the native Hawaiian bird fauna (Van Riper et al., 2002).

As has long been recognised (Gilpin and Soule, 1986), most extinctions are a result of multiple interacting factors. In some situations, infectious diseases may be responsible for a major decline in population numbers, and one or more other factors may be responsible for finally driving the now rare population to extinction. In other cases, a major decline may have occurred due to factors other than disease, but disease may finally drive the population to extinction. We, therefore, include drivers of major declines in our review, in addition to considering cases where disease itself may have been the final causative agent of extinction.

For the purposes of this review, we consider infectious diseases of species in the wild rather than infections of captive animals. Also, we focus on infectious diseases that, to the best of our knowledge, have caused declines and functional extinctions of species, regardless of whether they were proximately involved in the extinction of a species.

## Drivers of disease-associated declines

Whether a pathogen threatens a particular host depends on a complex interaction between the properties of the pathogen, characteristics of the host and the environment in which both exist, all considered in the context of the ecological community in which they are embedded and its evolutionary history. Figure 2 shows a conceptualisation for understanding the interaction of these factors, using amphibian chytridiomycosis as a case study.

Chytridiomycosis is caused by the fungus *Batrachochytrium dendrobatidis* (hereafter Bd) and is a fatal skin disease in amphibians. It is remarkable amongst wildlife pathogens for its devastating impact on global biodiversity. The fungus was first identified as the causative agent of multiple frog declines and extinctions in the Americas and Australia around the turn of the century (Berger et al., 1998; Longcore et al., 1999). Since then, studies of museum specimens and molecular tracing have revealed that it originated from the Korean peninsula and was spread around the world through amphibian trade over the last century (O'Hanlon et al., 2018). The fungus is maintained in aquatic environments by tolerant reservoir hosts (e.g., Figure 2b enabling it to cause more vertebrate extinctions (>70 species) and endangerments (≈500 species) than any other pathogen (Scheele

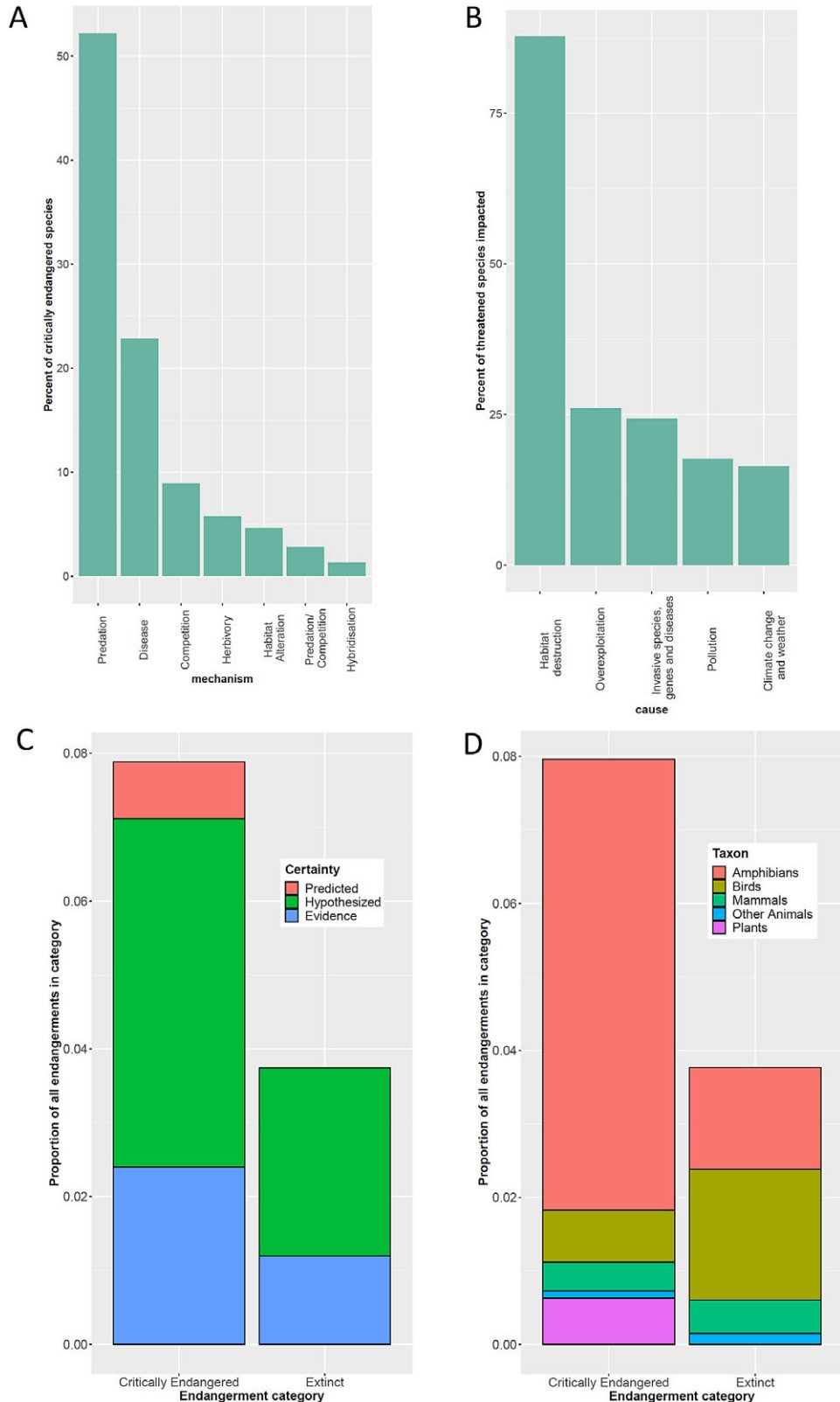

**Figure 1.** Extinctions or endangerments attributed to parasites and pathogens: (a) impact mechanisms of invasive species on vertebrates listed as critically endangered in the IUCN red list (Duenas et al., 2021; Supplementary Figure S2); (b) percentage of threatened species (vulnerable, endangered, or critically endangered) in the IUCN red list impacted by various drivers (Hogue and Breon, 2022), noting that infectious disease is included with "invasive species and genes", and that species may be impacted by more than one driver (thus the percentages add up to more than 100); (c) proportion of extinctions and critical endangerments in the IUCN red list attributed to infectious disease, relative to certainty (Smith et al., 2006); (d) proportion of extinctions and critical endangerments in the IUCN red list attributed to infectious disease, relative to taxonomic group (Smith et al., 2006).

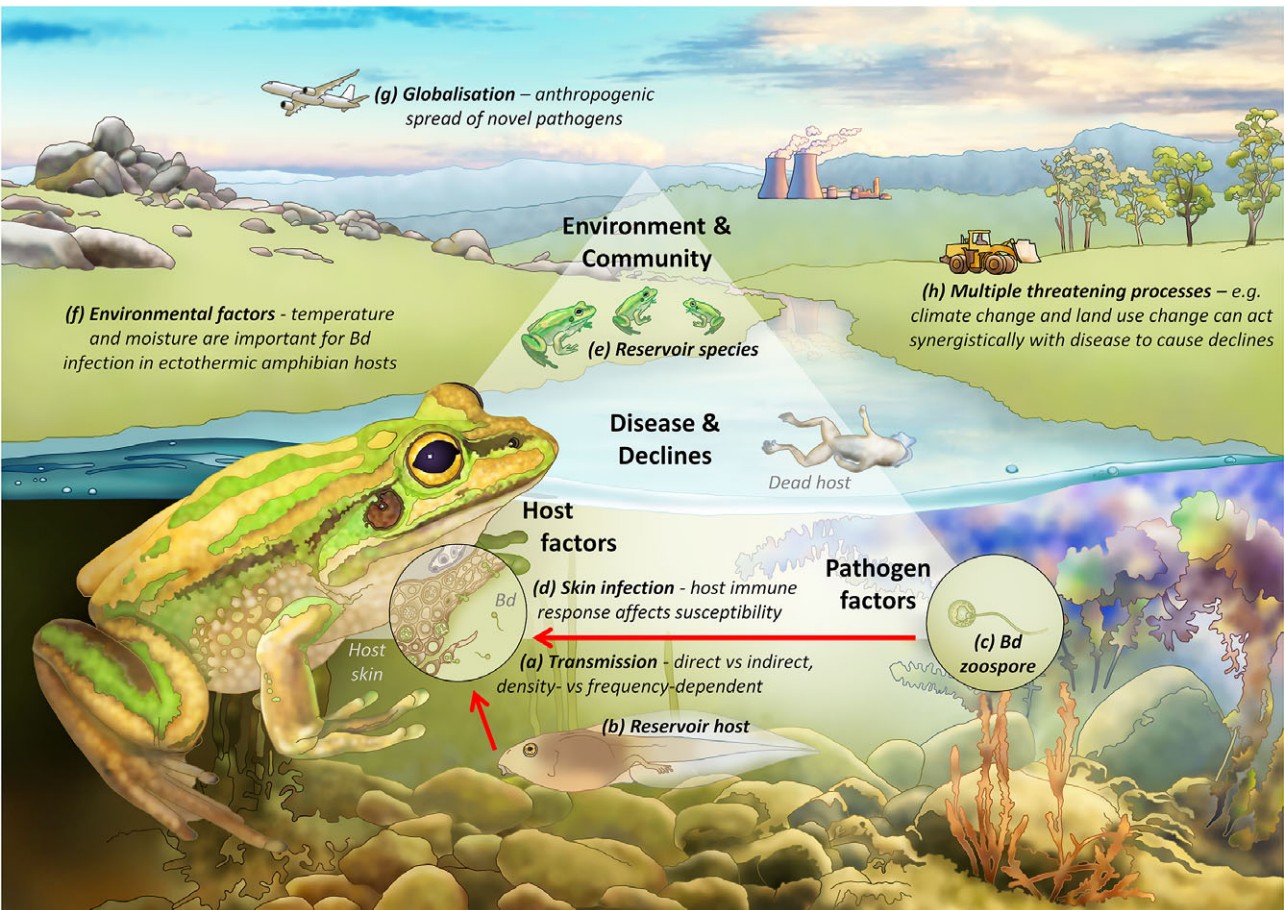

**Figure 2.** Conceptual framework for understanding disease-associated declines. Framed around the case study of chytridiomycosis in frogs, disease-associated declines are the result of an interaction between properties of the pathogen, characteristics of the host and the environment in which both exist, all considered within the context of the ecological community in which they are embedded and its evolutionary history. In this example, on the lower left the host of interest is represented as a green and golden bell frog (*Litoria aurea*), and on the lower right the pathogen is represented by the fungus *Batrachochytrium dendrobatidis* (Bd for short) in its infectious motile zoospore form. The interaction between host and pathogen, and the outcome of infection (represented as host mortality in the form of a dead frog), are influenced by the surrounding environment (aquatic systems) in the context of anthropogenic change (e.g., globalisation, climate change, habitat loss and fragmentation) and the amphibian community (e.g., population and community dynamics and the presence of reservoir hosts). Point (a) highlights that transmission can be both direct or indirect, and the transmission function can be either density- or frequency-dependent; point (b) highlights that tadpoles can act as a reservoir life-stage for Bd, harbouring the pathogen in their keratinised mouthparts; point (c) depicts the pathogen as the infectious motile zoospore stage; point (d) shows the process of infection within the skin of a frog, and highlights the role of host factors such as the immune response in determining susceptibility and infection outcomes; point (e) depicts tolerant reservoir species within the amphibian community; point (f) highlights that environmental factors such as temperature and moisture are particularly important for determining infection of ectothermic amphibian hosts; point (g) highlights that Bd was likely spread globally through anthropogenic means; and point (h) highlights that multiple threatening processes can act synergistically with disease in driving amphibian declines.

et al., 2019). However, there are now some signs of species' recovery (Jaynes et al., 2022).

### Factors related to infection transmission

In the case of obligate pathogens lacking an abiotic reservoir, transmission between hosts is a key limiting factor on their capacity to cause population declines. Simple models of a single host and pathogen often assume that transmission depends on the density of infected and susceptible hosts. This density-dependent transmission means that as host numbers decline, transmission will also decline, whereby, below a threshold host density, the pathogen will be lost from the system before the host itself becomes extinct (McCallum et al., 2001). However, in more complex situations, including cases of indirect transmission (e.g., Figure 2a shows that Bd is often transmitted indirectly via a water source), pathogen-induced extinction is a possibility for several reasons (de Castro and

Bolker, 2005). First, extinction is more likely when a highly susceptible species coexists with reservoir hosts – animals (species or life stages) that are able to tolerate infection. These reservoir hosts continue to produce infective stages even as the highly susceptible host declines towards extinction (Figure 2b shows the example of tadpoles as a common reservoir life-stage for Bd. Figure 2e shows sympatric amphibians more tolerant of Bd may also act as reservoirs). Second, transmission does not always depend on host density (McCallum et al., 2001; Begon et al., 2002). In sexually transmitted infections, or where vectors are involved, or when infective contacts depend on complex host behaviour (Herrera and Nunn, 2019), transmission may depend primarily on the frequency of infection rather than on host density. Frequency-dependent transmission does not require a minimum threshold of host density for infection maintenance, so extinction is possible. Third, if host density becomes sufficiently small, stochastic effects may result in host extinction due to infectious disease.

## Properties of pathogens

Certain properties of pathogens can make them more likely to cause population declines in their hosts. Viruses and prokaryotes tend to have short generation times and high mutation rates, which may enable them to rapidly increase virulence, jump host barriers and adapt to adverse conditions. However, their relative organismal simplicity means they also have certain weaknesses against the immunological defences of their hosts. In contrast, eukaryotes such as fungi, protozoans and metazoans share complex cellular machinery with their vertebrate hosts, making them more difficult to defend against. As an example, fungi are emerging as increasingly important pathogens of conservation significance (Fisher et al., 2012) and their impact is often strongly dependent on temperature, meaning that changes in their distribution and impact can be expected with climate change. Furthermore, many fungi have infective stages capable of persisting in the external environment for extended periods (e.g., the free-swimming infectious zoospore stage of Bd, Figure 2c), decoupling disease impact from population density.

Another important fungal disease is white-nose syndrome (WNS) of bats caused by *Pseudogymnoascus destructans*. The fungus attacks bats whilst they are hibernating and has caused local or regional extinction of a number of bat species in North America since its first detection in 2006 (Blehert et al., 2009). However, it appears to have coexisted with European bats for millennia (Hoyt et al., 2016). This pathogen has the capacity to survive off the host in an environmental reservoir within the caves where bats hibernate, meaning that transmission is not strongly dependent on bat population density. Whether or not populations can persist in the presence of the fungus appears to depend on a complex interplay between the environment within hibernacula (both humidity and temperature) and host traits (Grimaudo et al., 2022). WNS, therefore, illustrates both the importance of environmental reservoirs in pathogen threats and the importance of host traits in determining which species are particularly sensitive to extinction.

Sarcoptic mange is a skin disease of mammals caused by a metazoan – a mite *Sarcoptes scabiei* (Escobar et al., 2022). The species has a near worldwide distribution and has the ability to infect an exceptionally broad range of mammals. As a result, it can have devastating effects at a population level and has been responsible for the near extirpation of one of the densest wombat *Vombatus ursinus* populations in Tasmania (Martin et al., 2018). Sarcoptic mange is also having major effects on vicuna populations in South America (Monk et al., 2022) and threatens remnant populations of San Joaquin kit foxes (*Vulpes macrotis mutica*) in California (Rudd et al., 2020). Abbott (2006) speculated that mange may have been responsible for the collapse of the mammal fauna in Western Australia in the last decades of the 19th century. Evidence suggests that sarcoptic mange is increasing both in host and geographical range, although the reasons behind this are not clear (Escobar et al., 2022). Domestic species, particularly dogs, appear to play an important role.

## Host characteristics

Properties of hosts, both at the individual and population level, can make them particularly susceptible to infectious disease or certain pathogen groups (e.g., ectotherms and fungal diseases). It is widely assumed that low genetic diversity is associated with susceptibility to infectious disease, and this has indeed been demonstrated in several studies (e.g., lungworms in bighorn sheep (Luikart et al., 2008) and ectoparasites in hawks (Whiteman et al., 2006). An infectious cancer in which tumour cells are the infective agent, threatens the largest surviving marsupial carnivore, the Tasmanian devil *Sarcophilus harrisii*, with extinction (McCallum et al., 2009). Tasmanian devils have very low genetic diversity, including in the MHC complex, and it had been thought that this low genetic diversity was responsible for devils not recognising tumour cells as non-self (Siddle et al., 2007). However, devils are capable of mounting an immune response to the tumour (Caldwell and Siddle, 2017), and there has been rapid evolution in devil populations at loci related to disease resistance (Patton et al., 2020).

Host characteristics may also make them resistant or tolerant of infection. For example, amongst mammals, bats appear to have an unusual tolerance of viral pathogens, possibly because of the high metabolic rate required for flight (Brook and Dobson, 2015). This means that bats may be less threatened by viral pathogens, but it may also mean that they are potentially reservoir hosts for a range of viral pathogens affecting other species. By analogy, in the example of chytridiomycosis, a range of host factors have been demonstrated to affect susceptibility to infection and disease, including the presence of symbiotic bacteria and host immune mechanisms (Figure 2d shows a magnified view of the processes of infection within the skin of a frog).

## Environmental and community characteristics

The ecological community in which a host-pathogen interaction is embedded is critically important in determining whether a pathogen is capable of threatening the extinction of the focal host. If the community contains reservoir species capable of maintaining the pathogen with limited effect on survival, then there may be a high force of infection even as the focal host declines towards extinction. Chytridiomycosis and WNS are examples of diseases where biotic or abiotic reservoirs play an important role in driving declines (Figure 2e shows that multiple frog species often share the same aquatic environment and can act as infection reservoirs).

Unusual environmental conditions may permit pathogens that are normally relatively asymptomatic to produce mass mortality. For example, saiga antelope *Saiga tatarica* in central Asia suffer from occasional mass mortalities – up to 60% of the entire species population – caused by the bacterium *Pasteurella multocida* (Kock et al., 2018). The bacterium is usually present at high prevalence, but infections are essentially asymptomatic. Under conditions of high humidity and high temperatures, however, it can cause these mass mortalities. Similarly, in the case of chytridiomycosis, environmental temperature and moisture are important factors determining infection dynamics in ectothermic amphibian hosts (Figure 2f; Sasso et al., 2021).

As is evident from several of the examples we have given above, many extinctions due to infectious disease have occurred on islands (Wikelski et al., 2004). There are several reasons why this is the case. First, island populations are often small, which means that a factor such as disease that causes decline may lead to extinction due to stochastic factors (de Castro and Bolker, 2005). Second, island communities are prone to invasions. These may be pathogens themselves, vectors, or hosts capable of acting as reservoirs. Third, as is elaborated below, when hosts are exposed to pathogens of which they have no evolutionary experience, they are likely to be particularly severely affected, and the isolation of island species limits their previous exposure to many pathogens. Nevertheless, not all emerging pathogen threats are restricted to islands.

Chytridiomycosis and WNS are examples of pathogen threats that have occurred on a continental level.

### Emerging and novel pathogens

When hosts are exposed to novel pathogens, with which they have not evolved, effects may be especially severe (McCallum, 2012). The examples discussed above involve primarily situations where pathogens have recently arisen in the community or host. In most cases, such as avian malaria in Hawaii or chytrid fungus, the pathogen has been recently introduced from elsewhere, typically by human intervention (Figure 2g highlights that Bd was most likely spread around the world anthropogenically via globalisation, such as in amphibian trade or on inadvertent stowaways). In other cases, land use change, climate change or habitat destruction may act as synergistic threats and can bring highly susceptible hosts into contact with pathogens to which they previously have not been exposed (Figure 2h highlights the additive threat of global warming and habitat loss to frogs threatened by Bd). In rare cases, of which Tasmanian devil facial tumour is an example, the pathogen may be entirely novel, with no other cases known in any other species. When a novel pathogen first arrives, individual hosts will have no existing adaptive immunity, and at the population level, resistance or tolerance will not yet have evolved.

When a novel pathogen is first introduced into a population or community, dynamics are usually epidemic. This means that there is a rapid increase in prevalence of the pathogen, obvious mortality amongst the hosts and spatial spread from the point of introduction. These are hallmarks of a pathogen impacting a population (Preece et al., 2017). The effects of parasites and pathogens that are endemic in a population – that is, they are long established and relatively stable in the level of infection, are much harder to detect than those which are still increasing in prevalence or spatial spread. They may be a significant threatening process but their effects on population viability can be difficult to assess without experimental manipulation (Preece et al., 2017; Grogan et al., 2018).

### Anthropogenic changes that may lead to increased extinction risk from infectious disease

### Introductions

One of the principal effects of humans on ecological communities is the homogenisation of both flora and fauna through introductions, both deliberate and inadvertent (Williams et al., 2015). In the context of parasites and pathogens, this may mean the introduction of pathogens themselves, or the introduction of reservoir hosts, amplifying hosts, bridge hosts, or vectors. Whilst we have already discussed several examples, below we provide some key mechanisms.

### Climate change

A major effect of climate change is that it restructures ecological communities (Lajeunesse and Fourcade, 2022) because different species respond in different ways and at different rates to changes in climate. This will potentially bring species in contact with parasites and pathogens with which they have had no previous exposure over evolutionary time. As previously discussed, extinction risks are particularly high when susceptible hosts are exposed to novel pathogens. In mammals, Carlson et al. (2022) found that climate change is likely to create hotspots of novel contacts between species that previously have had separate distributions, with bats likely to play a major role in linking mammals with novel pathogens.

An example of climate change reassorting ecological communities and increasing pathogen threats is the change in altitudinal range of forests, mosquito vectors, avian malaria and susceptible Hawaiian birds. *Plasmodium* occurs primarily at low elevations, because transmission is temperature-dependent, ceasing at 13°C (LaPointe et al., 2010). This means that there is a narrow band of rainforest at high altitudes within which birds highly susceptible to malaria can persist. However, a 2 °C projected temperature increase will result in the 13 °C isotherm increasing in altitude beyond the current range of rainforest on the Island of Hawaii (Benning et al., 2002). Whilst in the very long run, one might expect that the rainforest would extend upwards following increasing temperatures, this will occur much more slowly than the response of mosquitoes and *Plasmodium* to the increased temperature.

### Habitat loss and fragmentation

Habitat loss and destruction are usually considered to be the principal drivers of extinctions. However, their impact on disease threats to endangered species is rather more nuanced. In general, small habitat patches tend to support impoverished parasite communities (Roca et al., 2009), although this may not necessarily translate into reduced disease threats. In some cases, encroachment of humans or domestic animals such as livestock or dogs can increase disease threats significantly, whereby pathogen spillover from human-disturbed habitats to natural habitats increases with the amount of habitat edge (Faust et al., 2018). For example, humans are increasingly encroaching on the habitat of the great apes, with evidence that human diseases are posing increasing risks to gorilla and chimpanzee populations (Chapman et al., 2005; Zimmerman et al., 2022). Domestic and feral dogs are a primary driver of outbreaks of canine distemper virus and substantial mortality in Serengeti lion populations, although a range of wildlife species may also be involved (Viana et al., 2015). In the short term, habitat destruction can reduce population size, but may increase population density as surviving animals crowd into remaining habitat patches. This can cause an increase in infectious disease in the remaining populations (Lebarbenchon et al., 2006). There is evidence that chronic stress in wild animals, such as can be expected from habitat fragmentation, can lead to immunosuppression and therefore increased susceptibility to parasites and pathogens (Norris and Evans, 2000). Whilst this has been suggested as the reason for chlamydial infection threatening urban and periurban koala populations in Australia, there is little direct evidence to support the idea (Grogan et al., 2018).

### Developing approaches to mitigate extinction threats posed by infectious diseases

Strategies to mitigate extinction risks from infectious diseases can be grouped into three broad categories – strategies to prevent arrival of known or suspected pathogens; strategies to eradicate pathogens once they have arrived; and strategies to minimise extinction risks when eradication of the parasite or pathogen is no longer a possibility (Foufopoulos et al., 2022).

### Preventing arrival of pathogens of known or suspected importance to conservation

Globalisation of trade and international travel have led to unprecedented introductions of pathogens to regions where they had been absent (Kilpatrick, 2011). Whilst it is impossible to forecast accurately which pathogens may cause major threats to biodiversity

if introduced into regions where they are currently absent, basic principles of biosecurity, such as quarantine, can reduce such risks. There are some cases where a pathogen is known to pose a serious threat, if it were to be introduced.

As described earlier, the amphibian chytrid fungus *Batrachochytrium dendrobatidis* (Bd) has caused waves of extinction of frogs globally. Melanesia has an intact frog fauna with high levels of species richness and endemicity and, as yet, no evidence of Bd infection (Dahl et al., 2012; Bower et al., 2019; Alabai et al., 2020; Oliver et al., 2022). Specific protocols at points of entry and monitoring for Bd are therefore justified. A related chytrid fungus *Batrachochytrium salamandrivorans* has recently emerged to threaten newts and salamanders in Europe (Martel et al., 2013; Martel et al., 2014). It was likely introduced from Asia. Should it be introduced to North America, a global hotspot for urodele biodiversity, there is a risk of multiple extinctions. The pet trade is a major potential route of introduction, and strict controls on import of newts and salamanders to North America are important (Grant et al., 2017).

### *Eradicating new invasions of pathogens or parasites with the potential to threaten extinction*

Global eradication of a pathogen is defined as driving the pathogen to extinction on a worldwide basis, whereas elimination is doing so within a defined geographical area. Global eradication has been successful for only two pathogens – Rinderpest in cattle (Njeumi et al., 2012) and smallpox in humans (Fenner, 1983). Due to the often cryptic nature of wildlife and their pathogens, it is highly unlikely that a pathogen of conservation significance would be able to be eradicated, even in the early stages of invasion or emergence. However, elimination of a pathogen from a wildlife population has been successful in some cases, such as the oral vaccination programme against rabies in foxes in Europe (Freuling et al., 2013). Developing a vaccine against a novel wildlife disease and delivering it successfully to wild animals at a rate sufficient to eliminate disease is difficult, and we know of no other case where this has been successful at eliminating a pathogen from a wild population. Culling, whether untargeted or of only infected animals, is problematical (Miguel et al., 2020), especially with endangered species, and we know of no case where it has successfully eliminated a disease that threatened the extinction of any wildlife population.

### *Managing populations of endangered species subject to disease threats that cannot be eliminated*

In practise, the best that can be hoped for in most situations is to employ strategies that enable an endangered species to continue to persist in the face of a threat from pathogens. In principle, such strategies may be designed to increase the resistance or tolerance of individual hosts to the pathogen or to reduce mortality from sources other than the pathogen so that the host population is better able to persist in the presence of the pathogen.

Vaccination to manage disease threats in wild populations has been attempted in many situations. Many vaccines require injection, necessitating capture and restraint of animals, and this can be impractical and may have substantial negative consequences (Burrows et al., 1995). However, for species of conservation significance, even if vaccines do not prevent infection, they may reduce

the severity of clinical disease. There is some evidence that this may be the case for vaccines against chlamydial infection in koalas (Waugh and Timms, 2020).

Pharmacological treatments for infection can be effective but are difficult to deliver to wild animals (Miguel et al., 2020; Wilkinson et al., 2022). There have been some intriguing examples where particular aspects of host behaviour can be manipulated to deliver drugs without the necessity of capturing wild animals. For example, we have already discussed the threat posed to wombat populations by mites causing scabies. As wombats are obligate burrowers, acaricide can be delivered to them via suitably designed flaps on burrow entrances (Wilkinson et al., 2022). Several species of Darwin's finches in the Galapagos are threatened by the introduced bott fly *Philornis downsi*. (O'Connor et al., 2014) The flies' larvae attack nestlings by burrowing into their flesh and pupate in nests. A possible method to control infestations is to make nesting materials impregnated with insecticide available to birds so that they incorporate them into their nests (Bueno et al., 2021).

Genetic management of populations offers the potential to increase resistance or tolerance to pathogens. There is evidence of rapid host evolutionary responses to pathogen threats in several of the examples we have already discussed, including Tasmanian devils and DFTD (Epstein et al., 2016; Patton et al., 2020), frogs and chytrid fungus (Voyles et al., 2018; Hollanders et al., 2022) and malaria in Hawaiian birds (Woodworth et al., 2005). If these evolutionary processes could be accelerated, it would provide a powerful tool for managing disease threats. The most obvious way to do this would be to select for (or engineer via targeted genetic intervention Kosch et al., 2022) resistance or tolerance and then release those genotypes into wild populations (e.g., Scheele et al., 2014). This would, however, be a complex and expensive undertaking. Alternatively, it might be possible to facilitate evolution of resistance or tolerance in the wild. This would require detailed knowledge of the ecological interactions in the community (see, e.g., Kilpatrick, 2006). As a first step, it is critical to ensure that other conservation actions, such as population supplementation or translocations, do not inhibit the evolution of resistance or tolerance in host populations (Hohenlohe et al., 2019).

## Conclusions

- Extinction threats posed by infectious diseases are probably under-recognised in the ecological literature.
- These threats are most often associated with anthropogenic influences and are likely to increase in the future.
- In some cases, evolutionary processes will lead to recovery after initial declines through the evolution of resistance or tolerance in the host or reduced virulence in the pathogen. It is important that management actions should not interfere with evolutionary processes.
- Proactive strategies need to be developed to recognise and manage these threats as early as possible, because once established, infectious diseases become very hard to manage.

**Open peer review.** To view the open peer review materials for this article, please visit http://doi.org/10.1017/ext.2024.1.

**Supplementary material.** The supplementary material for this article can be found at http://doi.org/10.1017/ext.2024.1.

**Acknowledgments.** This work was supported by Australian Research Council (ARC) grants DP180101415 (HM and LG) and DE200100490 (LG).

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
