## [Editor Report]

A review of your manuscript has now been received, with some minor revisions suggested before publication can proceed. I also see the topic discussed in this manuscript as one of interest to the readers of this journal, and in general I consider the manuscript to be well-written and reasonably thorough in its presentation of the relevant material. 

A general comment: The argument sometimes seems a bit muddled regarding the treatment of infectious disease as a driver of population decline or extinction, compared to it as the final event causing extinction after populations are reduced in size through other threatening processes. Perhaps some restructuring of the material could be done so that the distinction between these two mechanisms can be more clear. The “extinction vortex” concept is particularly relevant in this regard, where disease may deal the final blow to populations or species that have been reduced through habitat loss, genetic erosion, etc. The Partula example given in lines 67-71 is a prime example here. In contrast, the authors present the later example of disease contributing to population declines in saiga antelope (lines 205-207), without extinction as an explicit outcome. The manuscript could benefit from briefly comparing and contrasting these two mechanistic treatments of infectious disease in discussions of extinction.

Additional comments and suggested revisions are given below.

Lines 45-48: I suggest that the authors present a very clear definition of “infectious disease” that includes parasitic and fungal pathogens as “biological agents” in addition to more traditionally familiar bacterial and viral pathogens. This will improve clarity for those readers that may not be as familiar to the content presented here.

Line 56: It would be helpful here to clarify the definition of “technological limitations” -- for example limitations in our ability to detect the presence of the pathogen?

Line 108: I agree with the reviewer that Figure 2 does not add substantive information to the section beginning on line 104. Perhaps the authors can reassess this figure and revise it in a manner that more effectively compliments the accompanying text.

---

## [Editor Report]

I have reviewed your responses to the original set of reviews, and I very much appreciate your attention to improving the manuscript based on these suggested revisions. Overall, I believe the manuscript has been improved with these revisions, so publication of this manuscript can now proceed.

I extend my apologies for the lengthy delay in working this manuscript through the review and revision process. Thank you for your interest (and patience!) in publishing your manuscript in Cambridge Prisms: Extinction.